# Green Synthesis of Silver Nanoparticles Using *Diospyros malabarica* Fruit Extract and Assessments of Their Antimicrobial, Anticancer and Catalytic Reduction of 4-Nitrophenol (4-NP)

**DOI:** 10.3390/nano11081999

**Published:** 2021-08-04

**Authors:** Kaushik Kumar Bharadwaj, Bijuli Rabha, Siddhartha Pati, Bhabesh Kumar Choudhury, Tanmay Sarkar, Sonit Kumar Gogoi, Nayanjyoti Kakati, Debabrat Baishya, Zulhisyam Abdul Kari, Hisham Atan Edinur

**Affiliations:** 1Department of Bioengineering and Technology, Gauhati University Institute of Science and Technology, Guwahati 781014, Assam, India; kkbhrdwj01@gmail.com (K.K.B.); bijulipep@gmail.com (B.R.); nayanjyotikakati0@gmail.com (N.K.); 2SIAN Institute, Association for Biodiversity Conservation and Research (ABC), Balasore 756001, Odisha, India; patisiddhartha@gmail.com; 3Centre of Excellence, Khallikote University, Berhampur, Ganjam 761008, Odisha, India; 4Department of Chemistry, Gauhati University, Guwahati 781014, Assam, India; bkcsat@gmail.com (B.K.C.); skgogoi@gauhati.ac.in (S.K.G.); 5Malda Polytechnic, West Bengal State Council of Technical Education, Government of West Bengal, Malda 732102, West Bengal, India; tanmays468@gmail.com; 6Department of Food Technology and Biochemical Engineering, Jadavpur University, Kolkata 700032, West Bengal, India; 7Faculty of Agro Based Industry, Universiti Malaysia Kelantan, Jeli 17600, Kelantan, Malaysia; 8School of Health Sciences, Health Campus, Universiti Sains Malaysia, Kubang Kerian 16150, Kelantan, Malaysia

**Keywords:** silver nanoparticles, *Diospyros malabarica*, antibacterial, anticancer, catalyst, 4-nitrophenol

## Abstract

The green synthesis of silver nanoparticles (AgNPs) has currently been gaining wide applications in the medical field of nanomedicine. Green synthesis is one of the most effective procedures for the production of AgNPs. The *Diospyros malabarica* tree grown throughout India has been reported to have antioxidant and various therapeutic applications. In the context of this, we have investigated the fruit of *Diospyros malabarica* for the potential of forming AgNPs and analyzed its antibacterial and anticancer activity. We have developed a rapid, single-step, cost-effective and eco-friendly method for the synthesis of AgNPs using *Diospyros malabarica* aqueous fruit extract at room temperature. The AgNPs began to form just after the reaction was initiated. The formation and characterization of AgNPs were confirmed by UV-Vis spectrophotometry, XRD, FTIR, DLS, Zeta potential, FESEM, EDX, TEM and photoluminescence (PL) methods. The average size of AgNPs, in accordance with TEM results, was found to be 17.4 nm. The antibacterial activity of the silver nanoparticles against pathogenic microorganism strains of *Staphylococcus aureus* and *Escherichia coli* was confirmed by the well diffusion method and was found to inhibit the growth of the bacteria with an average zone of inhibition size of (8.4 ± 0.3 mm and 12.1 ± 0.5 mm) and (6.1 ± 0.7 mm and 13.1 ± 0.5 mm) at 500 and 1000 µg/mL concentrations of AgNPs, respectively. The anticancer effect of the AgNPs was confirmed by MTT assay using the U87-MG (human primary glioblastoma) cell line. The IC_50_ value was found to be 58.63 ± 5.74 μg/mL. The results showed that green synthesized AgNPs exhibited significant antimicrobial and anticancer potency. In addition, nitrophenols, which are regarded as priority pollutants by the United States Environmental Protection Agency (USEPA), can also be catalytically reduced to less toxic aminophenols by utilizing synthesized AgNPs. As a model reaction, AgNPs are employed as a catalyst in the reduction of 4-nitrophenol to 4-aminophenol, which is an intermediate for numerous analgesics and antipyretic drugs. Thus, the study is expected to help immensely in the pharmaceutical industries in developing antimicrobial drugs and/or as an anticancer drug, as well as in the cosmetic and food industries.

## 1. Introduction

Nowadays, nanotechnology is a rapidly developing field and has a wide range of applications in biomedicine, drug delivery, bioimaging, bio-sensing devices, optoelectronics, catalysis and also in environmental protection due to the exemplary properties, such as biocompatibility, high productivity, rapid production and cost-effectiveness [1,2,3,4,5]. Among nanomaterials, metal nanoparticles, such as silver, copper, zinc, gold, titanium and magnesium, have been gaining immense magnitude for their applications and play a major role in this field [6,7,8,9]. In recent years, silver nanoparticles (AgNPs) have created attention among researchers because of their important applications in biomedicine, agriculture, food industry, catalytic, biosensors, optoelectronics and optics [10]. Several physical and chemical methods have been widely employed for the synthesis of AgNPs, such as sonochemical, microwave, γ-rays, hydrothermal, wet chemical, laser ablation and sol-gel, but these methods have some disadvantages, such as their use of high beam energy, hazardous toxic wastes, require high capital costs and production of large amounts of toxic byproducts that cause environmental contamination [11,12,13]. Hence, there is a need to develop eco-friendly techniques without using toxic chemicals to overcome these limitations. Green synthesis of AgNPs is rapidly increasing due to its enhanced stability, nontoxicity, inexpensive, eco-friendly, simple and rapid method of preparation approaches, which are substitutes to hazardous physical and chemical methods. Green synthesized AgNPs from plants are being successfully employed in various pharmaceutical and biomedical fields, such as antimicrobial, antibiofilm, antifungal, anticancer, anti-angiogenic therapy, anti-inflammatory, antioxidant, antiviral, drug delivery systems, gene therapy, bioimaging and wound healing [14,15,16,17,18,19,20,21,22,23].

Among different organic pollutants in water, 4-nitrophenol (4-NP) is a toxic and harmful agent more often found in industrial and agricultural raw materials that pose negative impacts on the environment and human health [24,25]. It is important to remove such pollutants from water before any use, including domestic and industrial use. In the current study, it is observed that the researcher is devoted to converting the organic pollutants to less hazardous products in the aqueous medium. The 4-aminophenol (4-AP), a reduced product of 4-NP, has extensive use in the production of analgesic and antipyretic drugs. The 4-AP is also used as an anti-corrosion agent, hair dyeing agent or photographic developer. In general, the reduction of 4-NP to 4-AP is catalyzed by an iron/acid catalyst that poses major environmental impacts [26]. The use of corrosive acids in large-scale production can be achieved by using metal nanoparticles as a catalyst along with sodium borohydride (NaBH_4_). This method is found to be environmentally friendly. Despite thermal and electrical conduction, silver nanoparticles bear efficient catalytic properties. Silver nanoparticles can be synthesized using chemical, radiation, photochemical, Langmuir–Blodgettmethods, etc. [23]. However, green synthesis of silver nanostructures has received wide acceptance, as it possesses little hazard to the environment and human health [27]. Silver nanoparticles proved to have a unique increased ability to inhibit bacterial growth with no or negligible side effects when compared to antibiotics [28]. Cyclophanes synthesized AgNPs have shown wide applications in creating electrochemical and colorimetric sensors for the determination of heavy metal cations (Cu^2+^, Fe^3+^, Hg^2+^, Cd^2+^, Pb^2+^) anions (H_2_PO_4_^−^, I^−^), amino acids, polycyclic aromatic hydrocarbons and pesticides [29].

For the green synthesis of metal nanoparticles, natural products, such as extracts from various plants or parts of plants (leaves, fruits, flower, bark, peel, seed, root, latex) have received the most attention due to their growing success [30,31,32,33]. The plant extracts contain various phytocompounds that act as capping, as well as reducing agents for the synthesis of nanoparticles [34]. The antioxidant polyphenols and flavonoids are the major phytocompounds, mainly responsible for the formation of AgNPs by the reduction of silver ions into silver metal (Ag^+^ to Ag) and also act as a capping agent that stabilizes the size of the formed nanoparticles and shaped during the synthesis of nanoparticles. These phytocompounds also have a tendency to absorb on the surface of nanoparticles [35,36]. In the green synthesis of nanoparticles using plant extracts, nature, the concentration of the plant extract, metal salt, pH, temperature and incubation time during reaction affect the rate, production amount and properties of the formed nanoparticles [37]. Different plant extracts including *Azadirachta indica*, *Anthemis atropatana*, *Benincasa hispida*, *Bauhinia variegata*, *Catharanthus roseus*, *Caesalpinia pulcherrima*, *Coriandrum sativum*, *Melissa officinalis*, *Pedalium murex*, *Prosopis juliflora*, *Parkia speciosa*, *Stigmaphyllo novatum*, *Cotyledon orbiculata*, *Diospyros lotus* and Andean blackberry leaf have been successfully reported for the biosynthesis of AgNPs [15,16,32,38,39,40,41,42,43,44,45,46,47,48].

*Diospyros malabarica*, a species of flowering tree belonging to the family Ebenaceae, grows throughout India and other tropical regions of the world. The fruit of the *Diospyros malabarica* tree are round and yellow when ripe during the month of July and August. Additionally, *D. malabarica* has been reported to have various therapeutic applications [49]. This tree was also reported to be used in traditional medicinal practices in various diseases. Various bioactive compounds, such as gallic acid, flavonoid, anthocyanin, saponin, alkaloid, vitamin C, tannins, triterpenes, sitosterol and betulinic acid, are being reported to be present, which are mainly responsible for their antioxidant, antiprotozoal, antihelminthic, antiviral and anticancer activity [49,50]. We hypothesized that these phytoconstituents could be applied in the green synthesis of AgNPs. Considering the importance of the green synthesis of silver nanoparticles using different plants and the various medicinal properties of the *Diospyros malabarica* plant, it has been indicative for delving into the present study. This study focused on the green synthesis of AgNPs using *Diospyros malabarica* aqueous fruit extract as a reducing and stabilizing agent. The formed AgNPs were characterized by using UV-Visible spectroscopy, TEM, FESEM, EDX, XRD, FT-IR, DLS, PDI, zeta potential and photoluminescence (PL) properties. Furthermore, the antimicrobial activities of synthesized AgNPs were evaluated against human pathogenic microorganisms *Escherichia coli* and *Staphylococcus aureus*, as well as anticancer activity investigated on U87-MG (human primary glioblastoma) cell lines. The catalytic activity for the reduction of 4-nitrophenol was also tested for produced AgNPs.

## 2. Materials and Methods

### 2.1. Materials

Analytical grade silver nitrate (AgNO_3_) was procured from Sigma-Aldrich. Muller Hilton agar (MHA) for microbiology experiments was procured from HIMEDIA Laboratories (Mumbai, India). The bacterial pathogens *Escherichia coli* and *Staphylococcus aureus* used for the antimicrobial susceptibility evaluation were purchased from the Microbial Type Culture Collection (MTCC, Chandigarh, India). *Diospyros malabarica* fruit was freshly collected from the Morigaon district of Assam, India. Dulbecco’s modified Eagle’s medium (DMEM), fetal bovine serum (FBS) and MTT reagent (3-(4,5-dimethylthiazol-2-yl)-2,5-diphenyl tetrazolium bromide dye) was purchased from Sigma-Aldrich. The U87-MG (human primary glioblastoma) cell lines were obtained from the National Centre for Cell Science (NCCS), Pune, India.

### 2.2. Diospyros Malabarica (Aqueous) Fruit Extracts Preparation

The fruit of *Diospyros malabarica* was washed with tap water followed by distilled water until all the impurities were removed. Then, 20 g of fruit was weighed and immersed in 100 mL of double-distilled water (ddH_2_O) in an Erlenmeyer flask, were kept on a heated plate (60 °C) and allowed to boil for 15 min. The fruit extract was filtered by using Whatman filter paper no. 1. Pellets were removed, and the supernatant was collected in another tube. The filtrate (supernatant) was further kept refrigerated at 4 °C for the synthesis of AgNPs. 

### 2.3. Biosynthesis of AgNPs

For the green synthesis of AgNPs, silver nitrate AgNO_3_ (99.98%) reagent grade was used as a precursor. A total of 1 mM AgNO_3_ solution was prepared in ddH_2_O from a stock solution of 1 M AgNO_3_. Filtered aqueous fruit extract of *Diospyros malabarica* was mixed with this 1 mM AgNO_3_ solution in a 1:9 mL ratio in a conical flask and kept in the dark at 24 °C for incubation for 1 h. After the passage of time, the synthesis progress (reduction of Ag^+^ ions) was monitored by visual observation of change in color from light brown to dark brown. The color change indicates nanoparticle synthesis. Then, the green synthesized AgNPs were collected by centrifugation of 10,000 rpm for 15 min, and the precipitate was collected, washed with deionized water, dried and stored at room temperature. A schematic representation of AgNPs synthesis and its applications used in this study is depicted in Figure 1.

### 2.4. Characterization of AgNPs

#### 2.4.1. UV-Visible Spectroscopic Profile of Synthesized AgNPs

UV-Vis spectroscopy is the most important, simplest and most basic technique to confirm the formation of synthesized nanoparticles. UV-Vis spectra were recorded in the range between 300 and 700 nm using Cary 60 UV-Vis (Agilent Technologies, Santa Clara, CA, USA). In order to check the formation and stability of AgNPs, their SPR bands were recorded at different time intervals (0.5, 1, 1.5, 2, 2.5 and 4 h, respectively) during the green synthesis process. Double distilled water (ddH_2_O) was used as a blank to adjust the baseline.

#### 2.4.2. Stability of the Synthesized AgNPs

The stability of the green synthesized AgNPs from aqueous fruit extract of *Diospyros malabarica* in aqueous dispersions was determined by UV-Vis spectroscopy. The synthesized AgNPs were deliberately stored in a dark environment at ~30 °C. Afterward, the changes in the samples were monitored by UV-Vis spectra at 1 day and 60 days [51,52].

#### 2.4.3. Transmission Electron Microscopy (TEM)

TEM was applied to depict the morphology, size and particle size distribution of the green synthesized AgNPs. The images were taken by JEOL 2100 Electron Microscope (JEOL, Peabody, MA, USA). For TEM analysis, a single drop (10 µL) of AgNPs suspension was placed on the carbon-coated copper grid, dried for 2 h at room temperature (24 °C) and loaded into the specimen holder before performing analysis at 120 kV accelerating voltage.

#### 2.4.4. Field Emission Scanning Electron Microscope (FESEM) and Energy-Dispersive X-ray Spectroscopy (EDX)

To study the cell surface topography and characterize the elemental composition of synthesized AgNPs, FESEM combined with EDX were analyzed by using Sigma 300 (Carl Zeiss, Germany), operated at a 20 KV. A small amount of AgNPs (10 µL) was drop-cast on a clean coverslip and dried. The samples were then sputtered with gold and observed under a FESEM and EDX [53]. 

#### 2.4.5. PolydispersityIndex (PDI), Particle Size (DLS) and Zeta Potential (ζ)

The quantification of size (DLS), polydispersity index (PDI) and ζ-potential of the synthesized AgNPs was carried out using a Zetasizer Nano ZS90 (Malvern, UK). The PDI determines the spread of the particle size distribution. DLS analysis determines the hydrodynamic radius by approximating the particle size of the synthesized AgNPs. ζ-potential measures the surface charge of the synthesized AgNPs nanoparticles. 

#### 2.4.6. X-ray Diffraction (XRD)

XRD measurement was carried out by Phillips X’Pert Pro powder X-ray diffractometer (XRD) (PANalytical, Almelo, The Netherlands) with a copper target (CuKα1, λ = 1.54056 Å). It was operated with a nickel filter at a voltage of 40 kV and a current of 45 mA [54]. 

#### 2.4.7. Fourier Transform Infrared (FTIR)

To study the functional group present on the synthesized AgNPs surface, FTIR analysis was performed by using an FTIR spectrophotometer (Perkin Elmer spectrum 100 FTIR, 710 Bridgeport, CT, USA). The FT-IR spectra were scanned with wave numbers ranging between 4000 and 400 cm^−1^ at a resolution of 4 cm^−1^ in the transmittance mode [3]. 

#### 2.4.8. Photoluminescence (PL)

The PL spectra of biosynthesized silver nanoparticles were recorded by using the spectrofluorometer (Jasco, Oklahoma City, OK, USA, FP-8300).

#### 2.4.9. Antibacterial Activity of AgNPs

The antibacterial activities of the silver nanoparticles (AgNPs) were assessed against strains of clinical pathogenic microorganisms, *Escherichia coli* (*E. coli*) and *Staphylococcus aureus* (*S. aureus*), by using the agar well diffusion method [55]. Sterile and solidified Muller Hinton agar (20 mL) plates were swabbed with the microbes. Four wells were bored on each plate using a sterile well cutter, and all the experimental samples were given in triplicate in each plate. The wells were loaded with *Diospyros malabarica* aqueous fruit extracts (1000 µg/mL, 50 µL), synthesized AgNPs (500 and 1000 µg/mL, 50 µL) and double-distilled water (ddH_2_O) as the negative control (50 µL), respectively. Three antibiotics (Streptomycin 10 µg, Tetracycline 30 µg and Chloramphenicol 30 µg) were loaded and tested against *E. coli* and *S. aureus* in each plate as the positive control. After 24 h of incubation at 37 °C, the zone of inhibition was measured in mm for bacterial strains. The experiments were repeated thrice, and the diameter of the zone of inhibition was expressed as mean ± standard deviation (SD). 

#### 2.4.10. Assessment of Cytotoxicity in U87-MG Cells

The human cancer cell lines U87-MG (human primary glioblastoma) were grown as a monolayer in DMEM, supplemented with heat-inactivated 10% FBS and 1% antibiotic. The cell lines were maintained and grown at 37 °C with a humidified atmosphere containing 5% CO_2_ and 95% air. Both the cancerous cell lines (1 × 10^4^ cells/well) were seeded in a 96-well microtiter plate and incubated at 37 °C for 24 h. After incubation, various concentrations (µg/mL) of synthesized AgNPs were treated in triplicates and incubated at 37 °C for 48 h. Then, the cytotoxicity was checked by MTT cytotoxicity assay. The media of 96-well plates were discarded, followed by the addition of 100 µL of MTT dye (5 mg/mL in phosphate buffer saline (PBS; pH 7.4)) in each well and incubated at 37 °C for 4 h. After incubation, MTT was removed, and dimethyl sulfoxide (DMSO; 100 µL) was added to dissolve the formazan crystals formed due to reduced MTT, and the absorbance was recorded. A microplate reader SYNERGY-H1 (Biotek) was used to estimate the amount of reduced MTT by measuring the optical density (OD) at 570 nm with a reference filter of 655 nm [39]. The IC_50_ value was calculated by using the following equation:$$\mathrm{Cell}\;\mathrm{viability}\;\left(\%\right)=\frac{\mathrm{Absorbance}\;\mathrm{of}\;\mathrm{sample}}{\mathrm{Absorbace}\;\mathrm{of}\;\mathrm{control}}\times 100$$

##### Cell Morphology of AgNPs Treated U87-MG Cells

The morphology of the U87-MG cells after the treatment of synthesized AgNPs was studied under an inverted microscope. A total of 1 × 10^4^ cells were seeded in 96-well plates in DMEM along with 10% FBS and incubated for 24 h at 37 °C in a CO_2_ incubator. After incubation, U87-MG cells were again incubated for 24 h with the treatment of 1 mg/mL of synthesized AgNPs as treated and without AgNPs as untreated. The comparative effect on the morphology of the U87-MG cells of both treated and untreated was evaluated. 

#### 2.4.11. Live/Dead Assay

A live/dead assay was performed for the qualitative assessment of cell viability after treatment of synthesized AgNPs. The U87-MG cell lines at a density of 2 × 10^4^ cells were seeded in 96-well plates for 24 h. After incubation, cells are treated with AgNPs concentrations of 1 mg/mL. A blank (without treatment) was kept for control. After incubation, the cells were washed with PBS (pH 7.4) and stained with propidium iodide (PI) and thiazole orange (TO). The plates were then incubated for 45 min and observed using a confocal laser scanning microscope (CLSM) (TCS SP8, Leica, Germany).

#### 2.4.12. Catalytic Activity Test for Reduction of 4-Nitrophenol (4-NP)

To evaluate the catalytic efficiency of 4-NP by the AgNPs prepared through the mentioned green synthetic method, we have prepared a 50 mM solution of 4-NP. As a hydrogen source, we prepared a 0.1 M solution of NaBH_4_. When 15 mL of NaBH_4_ solution is added to 15 mL of 4-NP solution, the color of the 4-NP solution turns to light yellow from light green indicating the formation of 4-nitrophenolate ions. As a blank experiment, we added only NaBH_4_ to 4-NP and recorded UV-visible spectra at a regular time interval. To see the effect of catalyst AgNPs once we added both catalyst and NaBH_4_ together and recorded its UV-visible spectra. In the first experiment of catalytic activity, we have added 1 mL of AgNPs extract, while in the other, we have added 2 mL of catalyst solution to observe the effect of catalyst amount variation in the reduction of 4-NP. 

## 3. Results and Discussion

### 3.1. Biosynthesis and Characterization of AgNPs

#### 3.1.1. Color Change and UV-Visible Spectroscopic Profile of Synthesized AgNPs

An apparent change in the color of the reaction mixture from light brown to dark brown after 30 min of incubation at room temperature in dark conditions was noticed in Figure 1. Indeed, no transformation in color was observed in AgNO_3_ solution in the absence of plant extract under similar circumstances. The change in color was considered as an indication of AgNPs formation by the reduction of Ag^+^ ions [35,56]. The formation of AgNPs was further confirmed by UV-Vis spectral study, which was measured at different reaction times (0.5, 1, 1.5, 2, 2.5 and 4 h, respectively) displayed in Figure 2a. The kinetics of the reaction revealed that the reaction was slow at the beginning, up to 30 min from the start of the reaction. After 30 min the nucleation of the reaction was initiated very quickly, and the formation of AgNPs occurred. This was reflected in the appearance of the characteristic band of AgNPs at ~430 nm after 30 min of reaction time (Figure 2a). The reaction was allowed to continue further, and no significant change of the peak was observed in the intensity of Ag peak, which exhibits towards the completion of the reaction at ~2.5 h. The broad absorption peak at λ = 430 nm represents the characteristics of surface plasmon resonance (SPR) of spherical and aggregate AgNPs formation [11]. The stability of the green synthesized AgNPs with respect to time was checked by UV-Vis spectroscopy after 1 day and 60 days. The characteristic λ = 430 nm peak was found in the synthesized AgNPs, indicating the stability of the synthesized AgNPs (Figure 2b). Thus, from this result, we can conclude that the synthesized AgNPs showed high aqueous stability since a minor reduction in absorbance was observed at 430 nm. A similar type of analysis with almost the same range of results was obtained previously in recent studies of green synthesis of AgNPs using extracts of different plants, such as pomegranate leaves, *Benincasa hispida*, *Prosopis juliflora*, *Allium cepa*, *Parkia speciosa* and *Salvia hispanica* [15,36,39,57,58,59].This suggests that the phytochemicals present in *Diospyros malabarica* fruit extracted successfully act as reducing and capping agents.

#### 3.1.2. TEM

The size and morphology of green synthesized AgNPs were evaluated by TEM analysis. Figure 3a displays the SAED pattern of synthesized AgNPs, whereas Figure 3b demonstrates the spherical shape of the synthesized AgNPs and Figure 3c showed the particle size distribution of the AgNPs. The synthesized nanoparticles are polydisperse and range in size from 8 to 28 nm with an average size of 17.4 nm. The selected area electron diffraction (SAED) pattern displayed in Figure 3a of the AgNPs with bright spots indicated the polycrystalline nature of synthesized AgNPs, and each of their diffraction rings has been indexed to 111 and 220, the corresponding face-centered cubic (fcc)crystalline structure of metallic silver, which matches with the database of Joint Committee on Powder Diffraction Standards (JCPDS, No. 04-0783). The results of SAED were in congruence with the XRD pattern. Our analysis of SAED was almost similar to previously obtained results in some green synthesized AgNPs [11,60]. Our findings followed previous reports, where plant extract as a reducing agent was utilized in the synthesis of AgNPs, and almost similar results have been reported for AgNPs with the size of a nanoparticle ranging from 2 to 75 nm [61,62,63]. Due to this ultra-small size, shape and high stability, these synthesized AgNPs can be used in conjugating drugs and for targeting cancerous cells [56,64].

#### 3.1.3. FESEM and EDX Analysis

Morphological characteristics of green synthesized AgNPs were determined by FESEM techniques. FESEM images of synthesized AgNPs were predominantly found to show spherical morphology with an average size of 48.72 nm (Figure 4a). The results evidently display that synthesized AgNPs material contains small grain-like particles, which were agglomerated to form crystals with an almost uniform spherical shape with a smooth surface (Figure 4a). A similar type of findings was previously reported [65,66,67]. Figure 4b displayed the EDAX spectrum of synthesized AgNPs, which describes the elemental analysis of the material. Silver peaks are seen in the spectrum, which represents the presence of silver ions as an ingredient element in the synthesized AgNPs. The Ag peak showed a weight percentage of 17.25 and an atomic percentage of 4.21. Other peaks, such as O, Na, Al, Cl, K, were observed. These elements were originated from the biomolecules present in the *Diospyros malabarica* fruit extract that was bound to the surface of the AgNPs. Strong Si peaks were observed due to the use of a glass coverslip where the samples were loaded, whereas Au peaks are due to gold sputtering during FESEM analysis. Our finding results were almost in agreement with previous findings [45,68,69]. 

#### 3.1.4. Particle Size and Zeta Potential (ζ) Analysis

The synthesized AgNPs were subjected to DLS and Zeta potential (ζ) measurement techniques for the analysis of size, PDI and surface charge of the nanoparticles. Particle size analysis showed two peaks, one of 22.26 nm and another of 1.032 nm, respectively, with PDI 0.954 (Figure 5a). This showed the polydispersity nature of our synthesized AgNPs. The stability and surface charge of the AgNPs was determined by zeta potential. It showed that the synthesized AgNPs were negatively charged with a zeta potential (mV) of −22.3 mV (Figure 5b). The results exhibit that the surface of the synthesized AgNPs was negatively charged, thereby having good colloidal nature, as well as well dispersed in the medium due to strong repulsion among the particles to evade the agglomeration [38]. The size of the green synthesized AgNPs was found nearly similar between the TEM and DLS analysis. The higher negativity value of the zeta potential specified stability and well-dispersed behavior of nanoparticles [70]. Thus, our results prove the efficacy of components present in the *Diospyros malabarica* fruit extract as reducing and capping agents for the synthesis of AgNPs.

#### 3.1.5. XRD Analysis

The exact nature of the silver nanoparticles can be attained from XRD analysis. The XRD patterns of green synthesized AgNPs from fruit extract in Figure 6 showed Bragg’s model diffraction peaks at 2θ on 32.1° and 64.4°, respectively. The peaks corresponding to the 2θ = 32.1° and 64.4° depicted (111) and (220) lattice planes for silver confirmed the face-centered cubic (FCC) crystalline nature of the AgNPs. This result corroborated with the XRD analysis reported earlier [61,71,72,73]. The diffraction patterns showed good agreement with JCPDS (no. 04-0783). Thus, XRD patterns clearly showed the crystalline AgNPs formed by the complete reduction of Ag^+^ ions by the aqueous fruit extract of *Diospyros malabarica*. The other unassigned peak ensued the crystallization of silver nanoparticles along with the organic moieties or impurities that bound to the surface of nanoparticles [74,75,76]. The d spacing values were calculated from the theta values diffraction patterns using the following formula:d = λ /2 sin θ
where λ is the wavelength (0.154 nm) of X-rays, and θ is Bragg’s angle of diffraction. The d spacing values were found to be d = 0.28 and 0.14 nm, respectively, which also corroborated with the (111) and (220) planes for silver nanoparticles.

#### 3.1.6. FTIR Spectroscopy Analysis

FTIR spectroscopy analysis was conducted in order to determine the functional group of *Diospyros malabarica* fruit extract that was involved in the green synthesis of AgNPs as reducing and capping agents. Figure 7 displayed the FTIR spectral bands of the *Diospyros malabarica* fruit extract and synthesized AgNPs from the fruit extract. Sharp transmittance peaks were noticed at 3435, 2078, 1642, 1382 and 671 cm^−1^ in the synthesized AgNPs. As displayed in Figure 7, the FT-IR spectrum of *Diospyros malabarica* aqueous fruit extracts was closely similar to the FT-IR spectrum of the synthesized AgNPs, with a marginal shift in peaks. This similarity plainly indicates that some of the residual moieties of the phytochemicals present in the *Diospyros malabarica* fruit extract reside on the surface of the synthesized AgNPs. The FT-IR spectrum of *Diospyros malabarica* aqueous fruit extracts exhibits several absorption peaks at 3411, 2088, 1645, 1382 and 656 cm^−1^, which are associated with several functional groups. Sharp transmittance peaks were also noticed at 3435, 2078, 1642, 1382 and 671 cm^−1^ in the synthesized AgNPs. The FT-IR spectrum peaks evidently indicate the role of *Diospyros malabarica* aqueous fruit extracts as reducing and stabilizing agents. The strong occurrence of intense peaks at 3411 and 3435 cm^−1^ was an indication of O–H stretching vibration type of hydroxyl and amine (N–H) functional groups. The absorbance peaks located between 3000 and 3600 cm^−1^ were assigned to the stretching vibrations of hydroxyl groups (O–H) and amine (N–H) [2]. O–H stretching vibration was characteristic of polyphenols and N–H stretching was attributed to the presence of amino acids, peptides and proteins [22,77]. The phenolic group of compounds present in the fruit extract has been shown as powerful capping and reducing agents for the formation of AgNPs by reduction of silver nitrate. More intense peaks were observed at 2088 and 2078 cm^−1^, represented the presence of alkynes C≡C stretched vibration because of numerous secondary metabolites dissolved in the sample. The peaks at 1645 and 1642 cm^−1^ are attributed to the amide I band and –C=C– stretching vibration band. The 1382 cm^−1^ peak was attributed to the –C–N– stretching band as well as the amide I band of proteins in the fruit extract [45].The amide band I was associated with the stretch mode of the carbonyl group (C=O) united to the amide linkage. The peaks at 656 and 671 cm^−1^ were assigned to CH out of plane bending vibrations [56]. This amide I band might also be due to the presence of proteins from the fruit extract [39,65]. The proteins might be capped or coated around the synthesized AgNPs for their stability and to prevent agglomeration [56]. Thus from the FTIR investigation, it was evident that bioactive compounds, such as polyphenols, along with proteins present in the *Diospyros malabarica* fruit extracts play an important role in the synthesis of AgNPs. 

#### 3.1.7. Photoluminescence (PL) Studies

The photoluminescence (PL) of the synthesized AgNPs by *Diospyros malabarica* aqueous fruit extract was analyzed by fluorescence emission spectroscopy to analyze its optical property. The PL study showed the luminescence spectrum of the synthesized AgNPs at room temperature. The visible luminescence of AgNPs is due to the recombination of electron-hole pairs between *d*-band and *sp*-conduction above the Fermi level [78]. The synthesized AgNPs were dispersed in water, and the PL emission spectra were recorded at the different excitation wavelengths. The normalized fluorescence spectra of synthesized AgNPs were shown in Figure 8a. This illustrated that the AgNPs solution was irradiated at varieties of excitation wavelengths ranging from 320 to 430 nm, which showed different emission intensities centered at 470 nm. The fluorescence emission spectra were recorded at a fixed slit width of 5 nm. Thus, emission possesses an excitation wavelength-independent emission property. The literature reveals that the excitation independent properties are achieved by nanoparticles (NPs) due to doping or the presence of heteroatoms present on the surface of nanoparticles [79]. The normalized absorption and emission spectra of green synthesized AgNPs are shown in Figure 8b. It was observed that there was little overlap of absorption and emission spectra having large Stokes shift (Δλ = 249.05 nm, or Δν = 8884.15 cm^−1^) in comparison to some of the commercial fluorescent dyes, such as fluorescein (Δλ = 24 nm, or Δν = 938 cm^−1^) and Rhodamine 6G (Δλ = 24 nm, or Δν = 823 cm^−1^) [80]. Such a large stokes shift helps to reduce self-quenching that occurs from molecular self-absorption. This instead makes the material useful in practical applications. The PL-quantum yield (QY) of AgNPs was measured following a standard procedure [81]. A comparison was made for integrated photoluminescence intensities (excited at 360 nm) and absorbance values (at 330 nm) using quinine sulfate (in 0.1 M H_2_SO_4_) as a standard (Φ = 0.54). The synthesized AgNPs and quinine sulfate were diluted to different concentrations so that it gives an absorbance below 0.1 at 330 nm. Then the PL-QY was calculated using the following equation.
Φ_x_ = Φ_ST_ × (G_x_ × G_ST_) × (η_x_^2^/η_ST_^2^)

Here G is the gradient of the plot, η is the refractive index of the solvent, Φ is the quantum yield while X refers to AgNPs, and ST refers to quinine sulfate with a refractive index of 1.33. Thus, a moderate quantum yield (0.06%) was found for the synthesized AgNPs. The plot of integrated fluorescence intensity versus absorbance was depicted in Figure 8c. The AgNPs in aqueous solution emits blue light, which is confirmed by CIE 1931 chromaticity parameters (x, y) calculated using CIE 1931 app [82], as shown in Figure 8d. We found CIE parameters for synthesized AgNPs solution as x = 0.21 and y = 0.54 when photo-excited at 430 nm that receives color temperature 8087.76 K. The triangle in the chromaticity plot represents the emission of blue light by AgNPs on excitation, which is in agreement with the normalized fluorescence spectra shown in Figure 8a. Thus, the synthesized AgNPs with optical properties can be utilized for bioimaging processes [83]. 

### 3.2. Antibacterial Activity of AgNPs

In the present investigation, the antibacterial effect of green synthesized AgNPs was subjected to evaluation on human pathogenic microorganisms *Escherichia coli* (Gram-negative) and *Staphylococcus aureus* (Gram-positive) by the agar well diffusion method (Figure 9). The antibacterial activities of *Diospyros malabarica* aqueous fruit extracts were also studied. Double distilled water (ddH_2_O) was kept as a negative control and antibiotics (Streptomycin 10 µg, Tetracycline 30 µg and Chloramphenicol 30 µg) as a positive control. The zone of inhibition around the samples of individual bacterial cultures is shown in Figure 9a–h. Significant inhibition zone was observed in both the microbes on treatment with AgNPs. The zone of inhibition was measured to be 8.4 ± 0.3 and 12.1 ± 0.5 mm at 500 µg/mL and 1000 µg/mL concentrations of AgNPs, respectively, against *Staphylococcus aureus*. While in the case of *Escherichia coli*, it was measured to be 6.1 ± 0.7 and 13.1 ± 0.5 mm at a concentration of 500 and 1000 µg/mL AgNPs, respectively. Lesser antibacterial activity was observed in the case of *Diospyros malabarica* fruit extracts (*Escherichia coli*; 5 ± 0.5 mm) (*Staphylococcus aureus*; 7 ± 0.8 mm) in comparison to AgNPs in both the microorganisms. These results were almost similar to the previous report studied on the antibacterial activity of synthesized AgNPs using leaf extract of *Diospyrosmalabarica* [84]. On the other hand, the negative control did not exhibit any antibacterial activity. All the three standard antibiotics used as positive control were found to be moderate to highly sensitive against *E. coli* and *S. aureus*. The zone of inhibition against these two bacterial strains was recorded as Streptomycin 10 µg: 14 and 12 mm, Tetracycline 30 µg: 32 and 34 mm and Chloramphenicol 30 µg: 31 and 33 mm, respectively. It was clearly observed that the green synthesized AgNPs displayed concentration-dependent antibacterial activity. The zone of inhibition of synthesized AgNPs was almost comparable to streptomycin but smaller when compared to the other two antibiotics used. However, the as-synthesized AgNPs displayed a considerable antibacterial activity against both of the strains. This antibacterial activity of green synthesized AgNPs might be due to their penetration through the bacterial cell wall that caused the structural damage by interacting with sulfur and phosphorous-containing biomolecules. This has also generated free radicals, which in turn damage the bacterial membrane on contact with them [82]. The inhibitory zone on both of the microbes was accredited on treatment with aqueous fruit extracts of *Diospyros malabarica* due to the presence of phytochemicals, such as flavonoids, alkaloids and polyphenols. Whereas the antimicrobial effect of AgNO_3_ was due to inhibition of proteins by binding with the thiol groups and also by binding with DNA, thereby arresting replication in bacteria as reported in earlier findings [84]. Previously, several studies reported the strong antimicrobial activity of silver nanoparticles synthesized from plant extracts. These AgNPs effectively killed a wide range of pathogenic microorganisms, e.g., *E. coli*, *S. aureus* and *P. aeruginosa* [85,86]. Our synthesized AgNPs possessed comparatively higher antimicrobial activity, as reported earlier by other workers [36,37]. Based on this experiment, synthesized AgNPs were found to have significant antibacterial efficacy against *Escherichia coli* and *Staphylococcus aureus*. The antimicrobial properties of these synthesized AgNPs may thus be utilized for medical, cosmetic and food industries. 

### 3.3. Cytotoxic Effect of AgNPs on U87-MG Cells

To check the antiproliferative activity of the *Diospyros malabarica* fruit extract mediated green synthesized AgNPs, different concentrations (0, 10, 20, 40, 60, 80 and 100 µg/mL) of nanoparticles were added to U87-MG cancer cell lines and incubated for 48 h. The results were analyzed by MTT assay, which showed a decrease in viable cells with an increasing concentration of AgNPs (Figure 10). The IC_50_ value of AgNPs was calculated to be 58.63 ± 5.74 μg/mL. Similar results were obtained from silver nanoparticles for U87MG cells reported earlier [87,88]. Green synthesized AgNPs using *Artemisia turcomanica* leaf extract also showed an IC_50_ value close to 4.88 and 14.56 μg/mL for AGS and L-929 cells [89]. Biosynthesized AgNPs using *Cucumis prophetarum* aqueous leaf extract also showed antiproliferative activity against A549, MDA-MB-231, HepG2 and MCF-7 cell lines with IC_50_ values of AgNPs 105.8, 81.1, 94.2 and 65.6 μg/mL, respectively [71]. The green synthesized AgNPs were found to be toxic in only cancerous cell lines and have negligible toxicity in normal cell lines [90,91]. The AgNPs also reported being capable of reducing various cell lines, such as HeLa, HepG2, PC3 and Vero cells [92,93]. The AgNPs with enhanced cytotoxicity tend to have high cellular uptake and retention via endocytosis and escaping efflux mechanism [88]. Further, at the IC_50_ concentration of AgNPs, the morphology of the cells was observed under an inverted microscope (Figure 11). The untreated cells showed regular elongated cell structure while most of the cells appeared to be shrunken and formed round vesicular structures in the cells treated with AgNPs. The distorted cell morphology of cells treated with AgNPs may be attributed to the occurrence of apoptosis. 

### 3.4. Live/Dead Assay Analysis

To study the cell viability qualitative assessment of *Diospyros malabarica* fruit aqueous-extract-mediated green synthesized AgNPs, live/dead assays were performed on U87-MG cell lines (Figure 12). Thiazole orange (TO) and propidium iodide (PI) stains were selected as nucleic acid-binding dyes to observe live and dead cells under a confocal laser scanning microscope (CLSM). TO stains all live and dead cells as it is permeant to membrane, and PI stains only dead cells due to impermeability to live cells with intact cellular membranes. The images showed that the live cells had green color fluorescence (including dead cells), and dead cells had red color fluorescence. The untreated control cells revealed large numbers of live cells. On the other hand, the treated U87-MG cells with biosynthesized AgNPs depicted more apoptotic cells. The superimposed images divulged apprehension of the dead cells with a yellowish-orange and disfigured structure [3,94,95]. This result was in agreement with in vitro cytotoxicity assay (MTT analysis). The reduction of cell survival rate of U87-MG cells treated with AgNPs in MTT assay corroborated with the images obtained by LIVE/DEAD imaging that further aided in visualizing the live and dead cells treated by synthesized AgNPs.

### 3.5. Catalytic Reduction of 4-Nitrophenol (4-NP)

To see the catalytic efficiency of AgNPs, we used the reduction of 4-NP in the presence of NaBH_4_ as a model reaction. The overall schematic representation for the reduction of 4-nitrophenol(4-NP) to 4-aminophenol (4-AP) was displayed in Figure 13. On the addition of NaBH_4_ to 4-NP, it produced a4-nitrophenolate ion. The 4-NP has a characteristic absorption at λ_max_ = 317 nm, while the 4-nitrophenolate ion absorbs at λ_max_ = 400 nm. On reduction of 4-NP to 4-AP, there would be a reduction of the peak at λ_max_ = 400 nm and the gradual appearance of a new peak at λ_max_ = 300 nm [26]. When we added 15 mL of 4-NP (50 mM) and 15 mL of NaBH_4_ (0.1 M) and stirred continuously for one hour as a blank (control) experiment, we noticed the effect of only sodium borohydride (NaBH_4_) in the reduction of 4-NP to 4-AP. There was a negligible change observed in the peak of 4-nitrophenolate at λ_max_ = 400 nm, as depicted in Figure 14a. However, on the addition of 1 mL of catalyst (AgNPs solution), we observed a continuous decrease in the λ_max_ of 4-nitrophenolate with the gradual appearance of a peak at 298–300 nm [24,26], as shown in Figure 14b. It was observed that within 15 min the 4-NP solution converted almost completely into 4-AP. In another similar experiment, we added an increased amount of catalyst. On the addition of 2 mL of AgNPs solution, 4-NP completely reduced within 5 min, as shown in Figure 14c. A plot C/C_0_ versus time (in minutes) was plotted (Figure 14d) to compare the efficiency of NaBH_4_ alone and various amounts of the catalyst with NaBH_4_. 

To study the kinetics of the reaction, a plot of −ln (C/C_0_) was plotted as a function of time and was displayed in Figure 14e. The C/C_0_ values were measured from the reduction of λ_max_ = 400 nm. The linear fit of the plot offered R-square values (coefficient of determination) 0.994 and 0.995, close to unity for a catalyst amount of 1 and 2 mL respectively. This supports the pseudo-first-order mechanism for the reduction of 4-NP. The apparent rate constants for the degradation of 4-NP by a catalyst amount of 1 and 2 mL were 0.20023 and 0.61127 min^−1^, respectively, were calculated from the slopes. The appearance of the peak at 300 nm was an indication of the production of 4-AP. This investigation was further supported by the isosbestic points that emerged near 250, 270 and 325 nm that confirmed the 4-AP as the only product in the reaction [26].

## 4. Conclusions

In the present investigation, we have successfully green synthesized AgNPs using an aqueous extract of *Diospyros malabarica* fruit. The method of preparation was simple, rapid, cost-effective and eco-friendly. The synthesized AgNPs were stable and smaller in size. The biomolecules present in the fruit, as well as proteins, contribute as a capping and reducing agent, which attributes to the synthesis of AgNPs. The biosynthesized AgNPs showed a significant antimicrobial effect against human pathogenic bacteria *Escherichia coli* and *Staphylococcus aureus*. The synthesized AgNPs exhibited significant dose-dependent anticancer activity on U87-MG (human primary glioblastoma) cells. Further, the synthesized AgNPs also act as a catalyst in the formation of 4-aminophenol by the reduction of 4-nitrophenol. 4-aminophenol has numerous applications in pharmaceutical industries, such as the production of antipyretic and analgesic drugs. Thus, the synthesized AgNPs showed their potential for implementations in cosmetics, therapeutics and food industries. However, in vivo experimental verification is required for its safe utilization.

## Figures and Tables

**Figure 1 nanomaterials-11-01999-f001:**
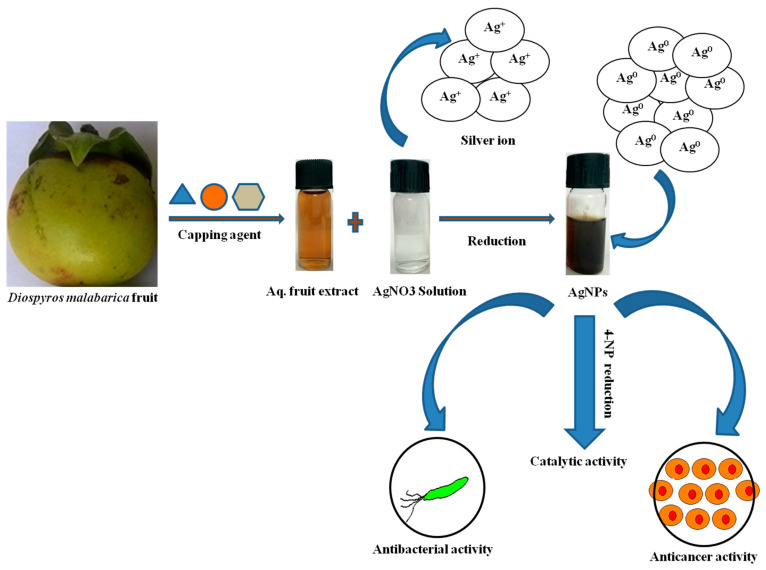
Schematic representation of the green synthesis of silver nanoparticles (AgNPs) and their applications.

**Figure 2 nanomaterials-11-01999-f002:**
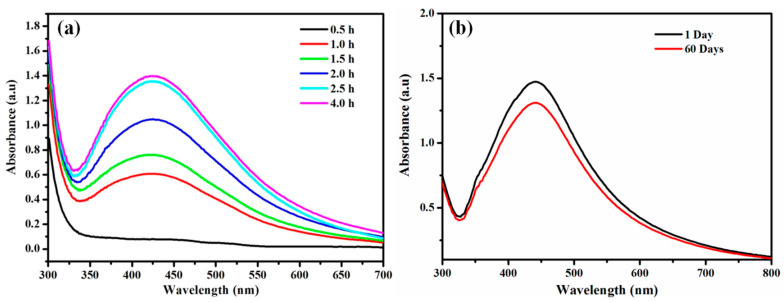
(**a**) UV-visible absorbance spectra kinetic reaction of the green synthesized AgNPs during the preparations at different time intervals. (**b**) UV-Vis spectra showing the stability of green synthesized AgNPs after 1 day and 60 days of preparation.

**Figure 3 nanomaterials-11-01999-f003:**
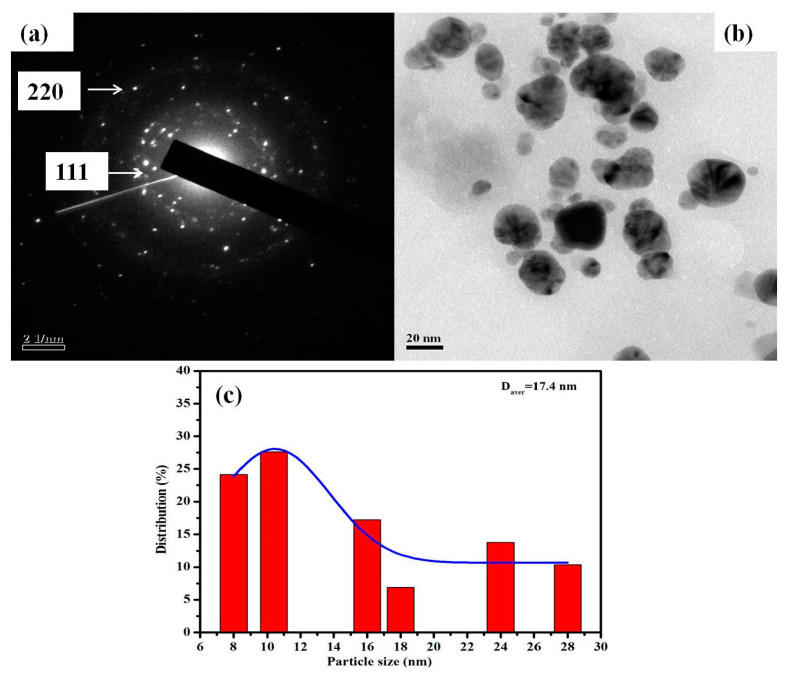
TEM micrograph of the synthesized AgNPs. (**a**) SAED pattern of synthesized AgNPs. (**b**) Analysis of the morphology of AgNPs. (**c**) Histogram showing the size distribution of AgNPs.

**Figure 4 nanomaterials-11-01999-f004:**
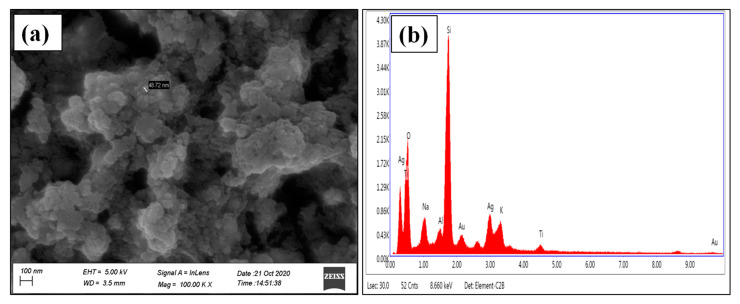
(**a**) FESEM recorded at the scale of 100 nm and (**b**) EDX image of synthesized AgNPs.

**Figure 5 nanomaterials-11-01999-f005:**
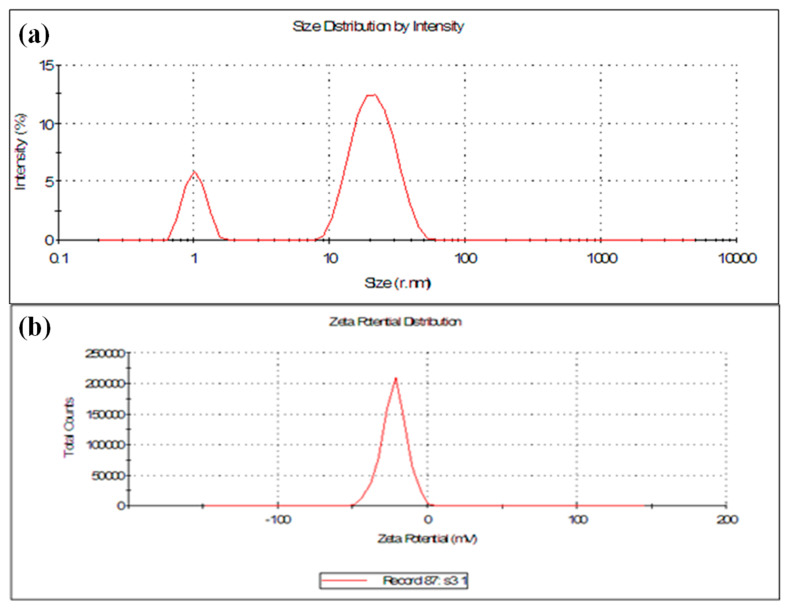
(**a**) Dynamic light scattering (DLS) and (**b**) zeta potential analysis of synthesized AgNPs.

**Figure 6 nanomaterials-11-01999-f006:**
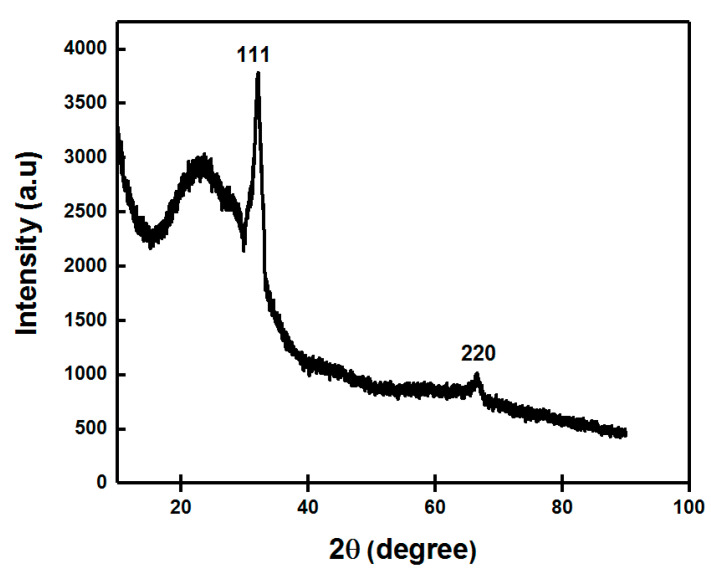
X-ray diffraction (XRD) pattern of green synthesized AgNPs.

**Figure 7 nanomaterials-11-01999-f007:**
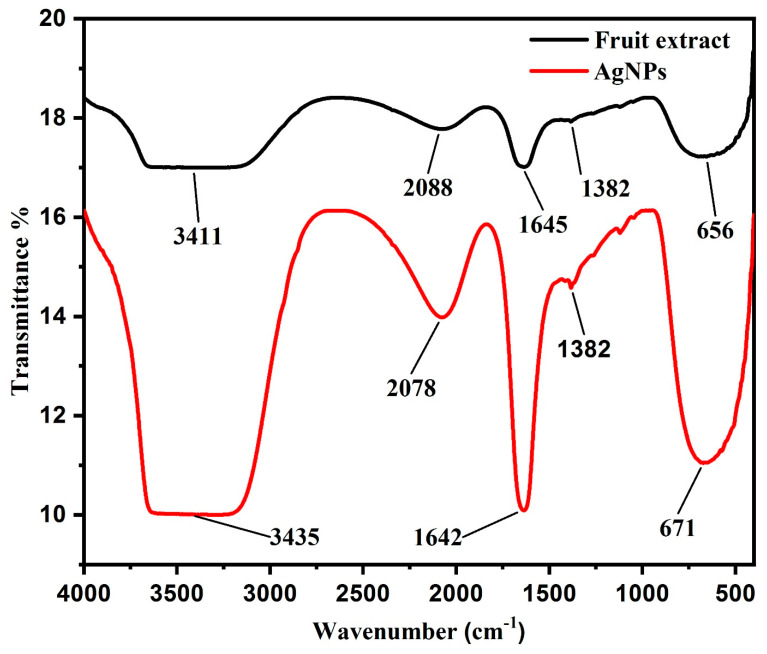
FTIR spectral analysis of *Diospyros malabarica* aqueous fruit extract and synthesized AgNPs.

**Figure 8 nanomaterials-11-01999-f008:**
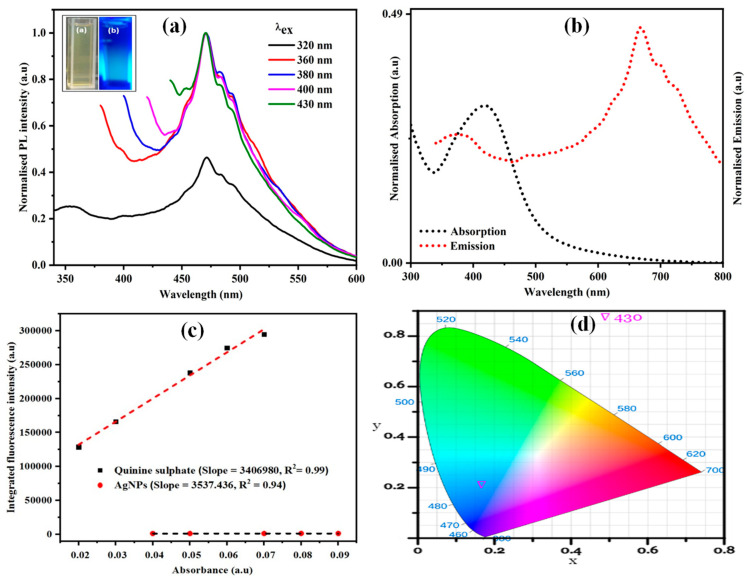
(**a**) Excitation-independent fluorescence emission spectra of green synthesized AgNPs (inset: AgNPs under (**a**) visible light, (**b**) UV light). (**b**) Absorption and emission spectra of green synthesized AgNPs. (**c**) Plot for calculation of quantum yield of AgNPs using quinine sulfate as standard. (**d**) Placement of the AgNPs fluorescence emission spectra on the CIE 1931 chromaticity chart.

**Figure 9 nanomaterials-11-01999-f009:**
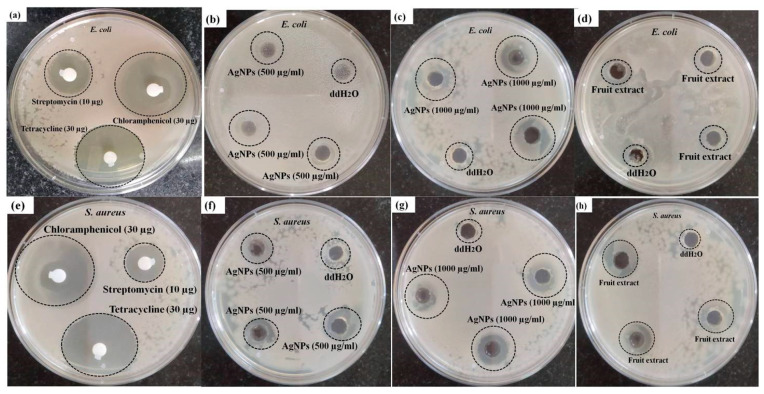
Antibacterial activities of (**a**,**e**) antibiotics (streptomycin(10 µg), Chloramphenicol (30 µg) and Tetracycline (30 µg)) as positive control, (**b**,**f**) synthesized AgNPs (500 µg/mL),(**c**,**g**) synthesized AgNPs (1000 µg/mL), (**d**,**h**) *Diospyros malabarica* aqueous fruit extracts (1000 µg/mL) and double distilled water (ddH_2_O) as negative control, on (**a**) Escherichia coli and (**b**) Staphylococcus aureus (all the experiments were performed in triplicates).

**Figure 10 nanomaterials-11-01999-f010:**
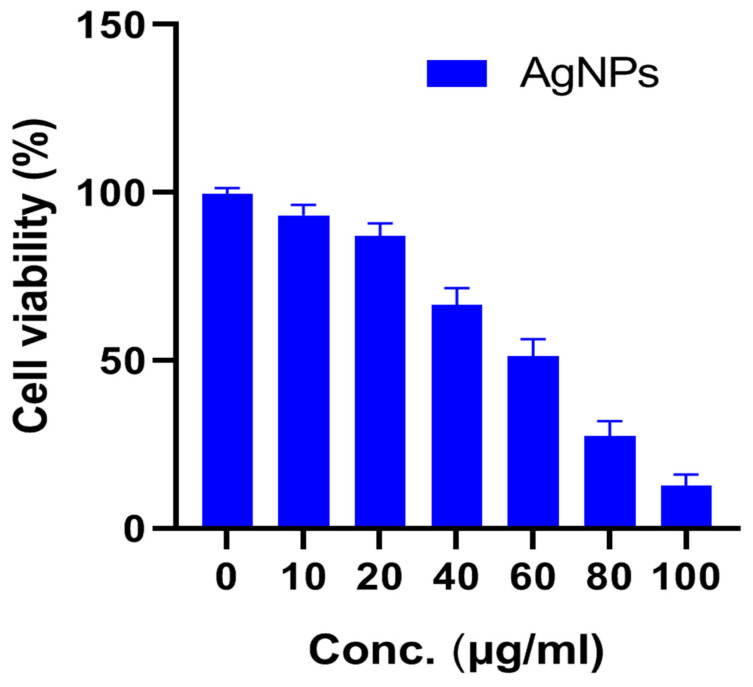
Effect of different concentrations of *Diospyros malabarica* extract mediated green synthesized AgNPs on the viability of U87-MG cell lines. Data represented as n = 3 ± S.D.

**Figure 11 nanomaterials-11-01999-f011:**
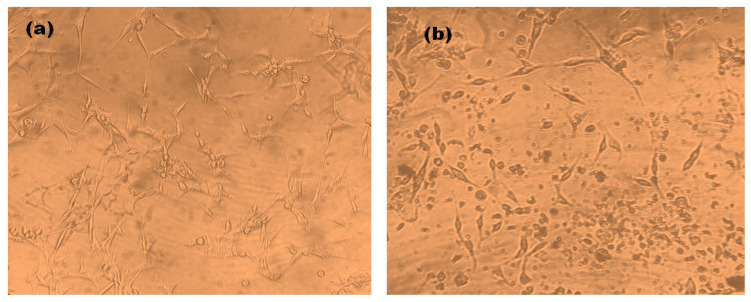
Cellular Morphology of U87-MG cells. (**a**) Untreated control, (**b**) *Diospyros malabarica* extract mediated green synthesized AgNPs treated at the IC_50_ concentration.

**Figure 12 nanomaterials-11-01999-f012:**
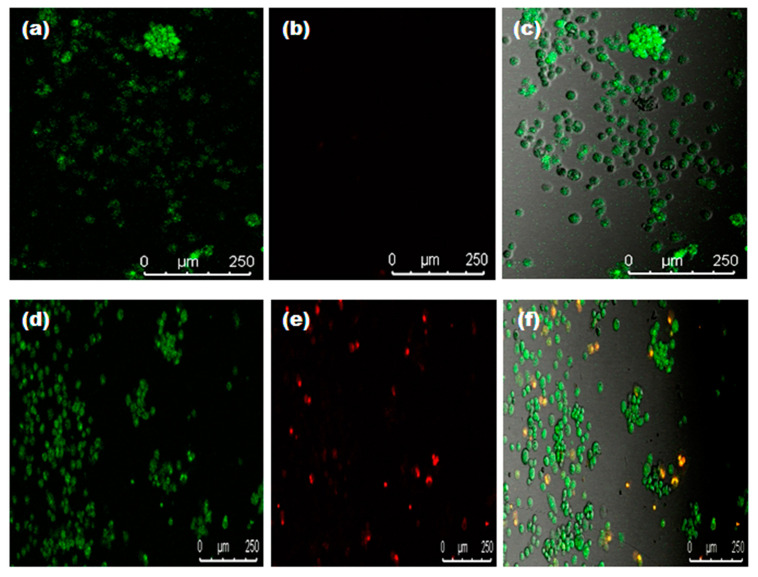
Confocal images of U87-MG cells at the IC_50_ concentration of *Diospyros malabarica* extract mediated green synthesized AgNPs incubated for 4 h. (**a**) TO, (**b**) PI, (**c**) superimposed image of untreated; (**d**) TO, (**e**) PI, (**f**) superimposed image treated with AgNPs.

**Figure 13 nanomaterials-11-01999-f013:**
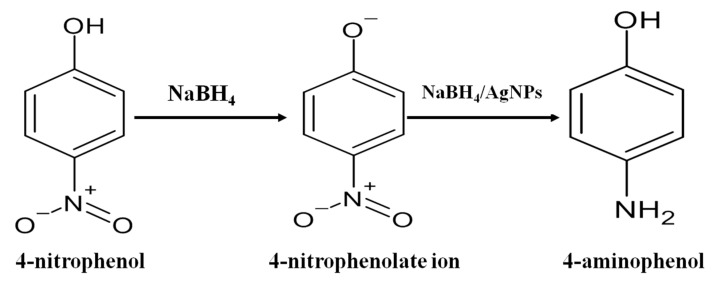
Schematic representation for reduction of 4-nitrophenol (4-NP) to 4-aminophenol (4-AP).

**Figure 14 nanomaterials-11-01999-f014:**
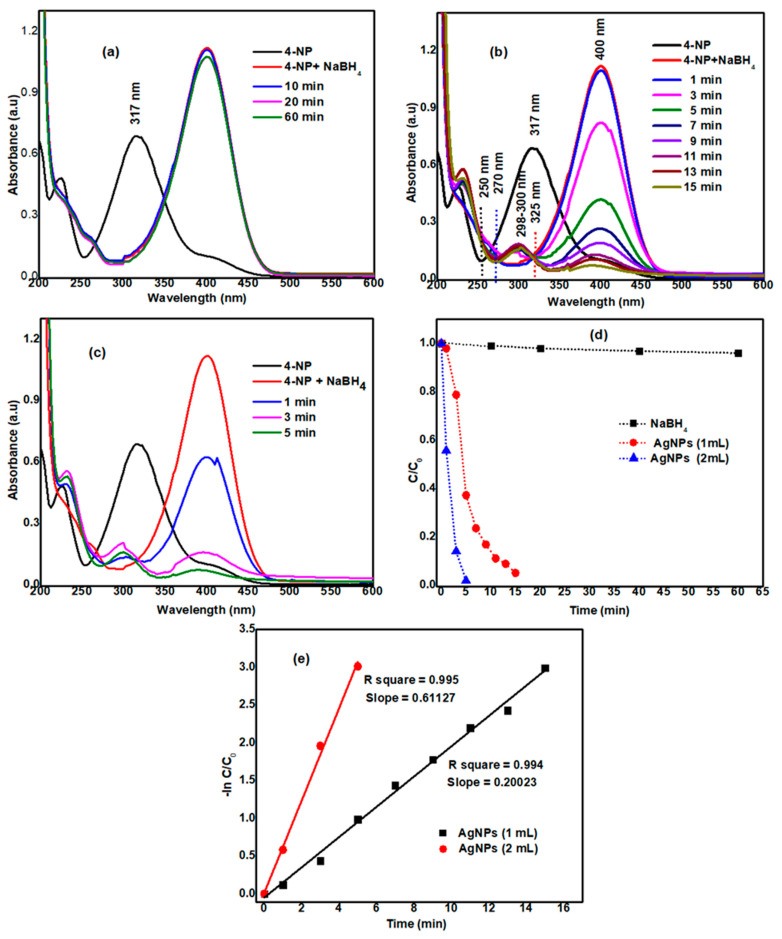
(**a**) Catalytic reduction of 4-NP by (**a**) NaBH_4_, (**b**) 1 mL AgNPs, (**c**) 2 mL AgNPs, (**d**) C/C_0_ and (**e**) −ln C/C_0_ vs. time.

## Data Availability

Not applicable.

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
