# Peer review of "Green Synthesis of Silver Nanoparticles Using Diospyros malabarica Fruit Extract and Assessments of Their Antimicrobial, Anticancer and Catalytic Reduction of 4-Nitrophenol (4-NP)"

_nanomaterials, 2021, doi:10.3390/nano11081999_

Round 1

Reviewer 1 Report

Basically, the approach taken in the study is interesting, and data may be appropriate for publication in Nanomaterials but after major revision due to some questions and comments: 

Why did the authors use only one ratio of filtered aqueous fruit extract of Diospyros malabarica to AgNO3 (1 : 9) during the synthesis of AgNPs?

Have other AgNO3 concentrations been tested for AgNPs synthesis?

In section Materials and methods, the authors mention 2 cell lines [HepG2 (human liver cancer) and U87-MG (human primary glioblastoma)] but present the results for one line only (U87-MG). Why were these cell lines used for research (actually one line)?

Why did the authors not do non-cancer line testing as a control? If something is recommended as an anti-cancer agent, it must be proven that it is not toxic to healthy cells.

For the synthesis of AgNPs, the authors used the fruit of Diospyros malabarica but did not perform a phytochemical analysis of this fruit. After all, depending on the ripeness of the fruit, its phytochemical composition changes.

Why did the authors not discuss the results obtained by other authors [Taranath et al. Phytosynthesis of Silver Nanoparticles Using the Leaf Extract of Diospyros malabarica (desr.) Kostel and its Antibacterial Activity Against Human Pathogenic Gram Negative Escherichia coli and Pseudomonas aeruginosa. Int. J. Pharm. Sci. Rev. Res., 30(2), January –February 2015; Article No. 18, Pages: 109-114. or Samaira et al., Green synthesis, characterization and photocatalytic applications of silver nanoparticles using Diospyros lotus. Green Processing and Synthesis, vol. 9, no. 1, 2020, pp. 87-96. https://doi.org/10.1515/gps-2020-0010], who also worked on the synthesis of AgNPs using Diospyros?

Why did the authors use one concentration of AgNPs in the antimicrobial tests? On what basis was this concentration selected? Where is the positive control - antibiotics? The publication cited at 44 did not investigate the antimicrobial activity, so it cannot be discussed here.

The correct record of bacteria is: Staphylococcus aureus, Escherichia coli and not Staphylococcus Aureus, Escherichia Coli.

The deletion in Fig. 1 is misleading because in my opinion it means no antibacterial and cytotoxic activity.

I do not agree with the last sentence in section conclusion “Thus, the synthesized AgNPs showed their excellent implementations and have promising applications in cosmetics, therapeutics, and food industries”, because the authors did not prove that the AgNPs obtained by them are safe.

Author Response

The authors of the manuscript are thankful to the reviewer for constructive suggestions and comments

Reviewer 2 Report

The manuscript nanomaterials-1300506 "Green synthesis of silver nanoparticles using Diospyros malabarica fruit extract and assessments of their antimicrobial, anticancer and catalytic reduction of 4-nitrophenol (4-NP)" by Bharadwaj et.al. describes the synthesis of silver nanoparticles using Diospyros malabarica fruit extracts and the study of their cytotoxicity, antimicrobial and catalytic activity. The synthesis of AgNPs was confirmed by UV-Vis, TEM, FESEM, EDX, XRD, FT-IR, DLS, Zeta Potential and Photoluminescence.

Questions and comments:

1) The authors should strengthen the Introduction part about silver nanoparticles. The authors should pay attention not only to methods of creating silver nanoparticles, but also to their applications. New articles on the design of silver nanoparticles, as well as their applications, should be added. For example, Int. J. Mol. Sci. 2020, 21(4), 1425; Appl. Sci. 2021, 11(3), 1120; Nanomaterials 2021, 11(5), 1343.

2) Part of Figure 2 was cropped. Figure 2 should be corrected.

3) The study of nanoparticles by DLS showed that the colloidal system is not monodisperse. How can the authors explain the high value of PDI (0.954)?

4) I suggest the authors compare IR spectra of synthesized AgNPs with IR spectra of Diospyros malabarica fruit extract.

5) The manuscript should be rechecked for typos and errors. For example:

- Line 161. … (10μ) …

- Line 239. … Ag+ ions.

- Lines 306–310. Different fonts

Author Response

(The authors gave the same response as above.)

Reviewer 3 Report

The work of Debabrat Baishya, Zulhisyam Abdul Kari and Hisham Atan Edinur et al. is devoted to a relevant topic in the field of designing nanomaterials as the green synthesis of AgNPs. The authors used the extract of Diospyros malabarica fruit to prepare AgNP. The resulting nanoparticles were studied as antibacterial and antitumor agents, as well as catalysts for the reduction of nitrophenols.

The work is a complex study with a large array of practical and theoretical material. The reliability of these results is beyond doubt. However, the reviewer has a number of comments / suggestions:

  1. In the introduction, the authors, in addition to the properties of Diospyros malabarica, must indicate the characteristics: the time of fruit ripening, how long it takes for the onset of fruiting.
  2. lines 308-309: Authors provide DLS values. However, judging by the presented results, the AgNPs system is polydisperse. For the PDI system, it was approximately equal to 1. For such systems, the zeta-potential measurement is not correct, since the analysis is carried out with a very large measurement error. It is not correct to draw any conclusions on such values of the zeta potential.
  3. line 351: In the FTIR spectra of extracts / AgNPs, the absorption band at 2354 cm-1 corresponds not to the -SH group of L-cysteine, but to CO2 dissolved in the extract. Therefore, the conclusions made further by the authors can be considered incorrect. Authors should conduct 1H NMR analysis of extracts where the -SH group is uniquely identified.
  4. lines 183 and 376: The authors should provide a more complete description of the fluorescence experiments (slit sizes, Stokes shifts, measurement time, possibly quantum yield, etc.). Without a detailed description of the experiment, the conclusions drawn by the authors are not correct.
  5. The introduction should add a reference as alternative methods of obtaining AgNPs: Shurpik, D. N.; Sevastyanov, D. A.; Zelenikhin, P. V.; Padnya, P. L.; Evtugyn, V. G.; Osin, Y. N.; Stoikov, I. I. Nanoparticles based on the zwitterionic pillar[5]arene and Ag+: synthesis, self-assembly and cytotoxicity in the human lung cancer cell line A549. Beilstein J. Nanotechnol. 2020, 11, 421-431, doi: 10.3762/bjnano.11.33

The work is of interest in the field of synthesis of silver nanoparticles and can be published after major revision.

Author Response

(The authors gave the same response as above.)

Reviewer 4 Report

This manuscript describes the development of silver nanoparticles (AgNPs) using Diospyros malabarica fruit extract. Further, the obtained nanoparticles were studied for their anticancer, antibacterial and catalytic activity.

In my opinion, this manuscript lacks of novelty since there are many works in the literature describing similar studies as the authors mentioned. Therefore, the authors should explicitly clarify the advantages of these nanoparticles regarding their applications.

Additional comments are as followed.

  1. The composition of the Diospyros malabarica fruit extract should be analyzed e.g. by Gas Chromatography/Mass Spectroscopy in order to identify its components.
  2. FTIR assignment is not correct, e.g. the band at ~2350 cm-1 corresponds to CO2 of atmosphere and not to SH of cysteine groups (stretching band of SH group of cysteine usually observed at 1550 cm-1, the band at 1644 cm-1 could only be attributed to Amide I and not to carbonyl groups of aldehydes, ketones or carboxyl groups (stretching band of C=O of aldehydes, ketones or carboxyl groups usually observed in the region between 1665 to 1730 cm-1). Please correct and modify the conclusions accordingly. The FTIR should be recorded again with higher resolution and after drying of the nanoparticles.
  3. The aqueous stability of the nanoparticles vs time should be assessed.
  4. The author claimed that the cell death mechanism of AgNPs attributed to apoptosis and in order to investigate this mechanism, they performed a Live/Dead Assay using thiazole orange and propidium iodide stains. This is not correct, since using this assay, only live or dead cells can be detected and it is impossible to conclude if the cell death is due to apoptosis. For this purpose, various cellular assays including Annexin V, Caspase and TUNEL detection methods, could be used. Therefore, correct the text accordingly and perform other assay to define the mechanism of action.
  5. Regarding the antibacterial activity of these nanoparticles, EC50 or MIC should be measured.
  6. The mechanism of the anticancer and antibacterial activity of these nanoparticles should be investigated in depth. Therefore, additional experiments are needed in order to study these activities, e.g. their intracellular uptake, cell cycle analysis, ROS generation, etc.
  7. The cytotoxicity assessment performed only on U87-MG cells and not on HepG2 cells. Please correct the text (lines 133 and 196), accordingly.

Author Response

(The authors gave the same response as above.)

Round 2

Reviewer 1 Report

  There are no comments.

Author Response

Thank you so much for accepting our manuscript.

Reviewer 2 Report

I thank the authors for answering my questions and improving the manuscript.

The quality of many of Figures is poor compared to the 1st version of the manuscript. I hope this is due to the conversion to pdf. Please, check it.

Author Response

Thank you so much for your comments and suggestion. We have enhanced the quality of the manuscript.

Reviewer 3 Report

Dear researchers Debabrat Baishya, Zulhisyam Abdul Kari and Hisham Atan Edinur et al. answered all the comments correctly and completely.

The manuscript can be Accept in present form.

Author Response

The authors are grateful to you for accepting the manuscript.

Thank You

Reviewer 4 Report

  1. FTIR analysis is not fully correct. Specifically, the band at 1642-1645 cm-1 is not only attributed to Amide I but also to –C=C- stretching vibration band. The band at 1382-1385 cm-1 is not attributed to Amide II but to -C-C-, -C-N- stretching band. Please correct the text accordingly.
  2. I have asked to assess the aqueous stability of the nanoparticles vs time. The authors answered me that they have performed this experiment and the results presented in Figure 2. But these results related to the kinetic of the NPs synthesis which is completed within 4h and not to the stability of the NPs. The stability vs time (for at least 1 month) could be measured by UV-vis or DLS since it is crucial parameter for these applications.
  3. The EC50 or MIC values of the NPs were not measured. At least comment the antibacterial results of NPs in comparison with those of the antibiotics.
  4. Last sentence of section 3.3 “The distorted cell morphology of cells treated with AgNPs may demonstrate the occurrence of apoptosis”. This cannot be concluded from the morphology of cells. Other experiments are needed to verify the apoptotic mechanism, otherwise modify the text by writing “….may demonstrate the occurrence of apoptosis” or “could be attributed to the apoptotic mechanism” or something like that.
  5. Write a short discuss for the results from the Live/Dead Assay in comparison with those from the MTT assay. The  images of Figure 12d-f 1 have poor resolution and their quality is low.

Author Response

The authors would like to thank the reviewer for your effort to share constructive comments on the manuscript. We have incorporated the changes as per the suggestions in the re-revised manuscript.

Round 3

Reviewer 4 Report

The manuscript can be accepted after minor revision since most of the comments have been addressed. Specifically, experimental details regarding the stability test by UV-vis spectroscopy should be added in the text. Additionally, in the discussion regarding the stability of NPs, the authors should mention that NPs showed high aqueous stability since a minor reduction in absorbance was observed at 430 nm.

Author Response

Thank you for considering our manuscript for publication, and also grateful for your constructive suggestions. The experimental details regarding the stability test by UV-Vis spectroscopy have been incorporated in the materials and methods section of the re-revised manuscript. As suggested, we have also mentioned the stability of the nanoparticles in the discussion section. All the changes are highlighted in yellow.

The supplementary figure 1 is now included in the main text as Figure 2b. Figure 2 has now changed to Figure 2a.
